Control of Cydia pomonella (L.) (Lepidoptera: Tortricidae) in apple orchards using the mating disruption technique

Kutalmış Alperen
Ögür Ekrem ekremogur@selcuk.edu.tr
Department of Plant Protection, Faculty of Agriculture, Selcuk University , Konya , Turkey
Hussein Mona
Electronic publication date: 2025 Oct 23
Publication date: 2025
Volume: 13
Electronic Location ID: e20226
Received 2025 Jul 11; Accepted 2025 Sep 22
Copyright: ©2025 Kutalmış and Ögür
Copyright year: 2025
Copyright holder: Kutalmış and Ögür
License: This is an open access article distributed under the terms of the Creative Commons Attribution License, which permits unrestricted use, distribution, reproduction and adaptation in any medium and for any purpose provided that it is properly attributed. For attribution, the original author(s), title, publication source (PeerJ) and either DOI or URL of the article must be cited.
License URL: https://creativecommons.org/licenses/by/4.0/

Keywords: Apple, Codling moth, Cydia pomonella, ISOCOD-C, Mating disruption, Türkiye

Funding: The authors received no funding for this work.

==============================
The codling moth, Cydia pomonella (L.) (Lepidoptera: Tortricidae), is a major, economically important pest of apple orchards in Türkiye. This study was conducted with the objective of evaluating the efficacy of the mating disruption technique in controlling C. pomonella in commercial apple orchards in the Beyşehir district (Konya) during the years 2023 and 2024. The experiments were conducted in six commercial apple orchards. Three of these orchards were treated with pheromones, while the remaining three served as control orchards. The efficacy of mating disruption was evaluated by comparing the number of C. pomonella males caught in Delta traps in pheromone-treated and control orchards and the infestation rates in these orchards. Delta traps baited with synthetic sex pheromone were hung in each pheromone-treated and control apple orchard to monitor the adult codling moths, and the number of males was recorded weekly. Once the first adult was caught in Delta traps, ISOCOD-C (380 mg (E,E)-8,10-Dodecadienol, dodecanol, tetradecanol) pheromone dispensers were hung at a dose of 500 pieces/ha, 1.5–1.8 m above the soil surface in four directions of the trees in the apple orchards where the mating disruption technique was applied. To determine the infestation rate of C. pomonella, 10 fruits from 10 trees (a total of 100 fruits) were randomly selected and the infested fruits were recorded weekly. ISOCOD-C pheromone dispensers suppressed capture of male moths in Delta traps and infestation rate in fruits in the treated orchards in both years, and the differences were found to be statistically significant in comparison to the control. In the pheromone-treated orchards, the mean number of males (trap/week) was 0.91 ± 0.18 and 0.81 ± 0.19 in 2023 and 2024, respectively, while this was 11.38 ± 1.64 in 2023 and 19.60 ± 2.65 in 2024 in the control orchards. The mean infestation rate (%) in the pheromone-treated orchards was 1.18 ± 0.21% and 2.50 ± 0.43%, in 2023 and 2024, respectively. In contrast, this rate was 13.26 ± 1.08% and 15.33 ± 1.02% in the control orchards. In addition, it was determined that the total number of sprays for codling moth in orchards using mating disruption decreased by 44.4% and 45.4% in 2023 and 2024, respectively, in comparison with the control. As a result, this study revealed that the ISOCOD-C pheromone disperser can be successfully used against C. pomonella in apple orchards.

Introduction

The apple, Malus domestica Borkh. (Rosaceae), is one of the most common and preferred fruits worldwide. It has great importance in terms of both its nutritional properties and its economic contributions. Apples are a rich source of health-benefiting nutrients, including fiber, vitamins A and C, potassium, flavonoids, antioxidant phenolic compounds, and minerals. These nutrients are known to reduce the risk of several forms of cancer, cardiovascular diseases, diabetes and asthma (Hyson, 2011; Skinner et al., 2018; Koutsos et al., 2020; Zhu et al., 2021; Asma et al., 2023). Global apple production has increased substantially over the past three decades, with a more than twofold rise from 56.8 million tons (t) in 1993 to 146.9 million t in 2023. China is the leading producer, with a yield of 49.6 million t, followed by the USA with 5.1 million t and Türkiye with 4.6 million t (FAO, 2025).

The codling moth, Cydia pomonella (L.) (Lepidoptera: Tortricidae), is one of the most devastating pests of apple orchards on a global scale, causing serious yield losses (Jaffe, Guédot & Landolt, 2018; Perrin et al., 2024; Wang et al., 2025). After hatching, the larvae of the pest penetrate the fruit in a short time, feed on the fleshy parts of the fruit and the fruit core for several weeks, and cause damage to the fruit by leaving droppings (Akroute et al., 2023; Erler & Tosun, 2023). The damage can cause up to 60% or even 100% crop loss in the absence of control measures (Wan et al., 2019; Ulaşlı& Can, 2024). In addition to apples, more than 30 fruit varieties such as walnut, pear, hawthorn, and pomegranate are among the hosts of the pest (Cheng et al., 2017; Çelik & Ünlü, 2017).

The most common control method used by producers in both Türkiye and other apple producing countries of the world to control C. pomonella is chemical control (Soleno et al., 2012; Fuentes-Contreras et al., 2014; Ju et al., 2021; Yeşilırmak & Ay, 2023). However, there are studies reporting that the codling moth has developed resistance to several classes of insecticides that are overused in its control (Reyes et al., 2007; Ioriatti et al., 2007; İşci & Ay, 2017; Bosch, Rodríguez & Avilla, 2018; Ju et al., 2021; Yeşilırmak, Çevik & Ay, 2025). In addition to resistance development, this method is also recognized to cause residue issues and have adverse effects on non-target organisms, beneficial insects, human health, and the environment (Geiger et al., 2010; Nicolopoulou-Stamati et al., 2016; Rani et al., 2021; Serrão et al., 2022; Khan et al., 2023). Therefore, research in recent years has focused on control methods that can be used as an alternative to chemical control. Among these, ‘biotechnical control’ is one of the most studied methods and appears as a viable alternative.

The mating disruption technique, which uses synthetically produced sex attractant pheromones to reduce insects’ ability to find mates, has proven to be an effective method for suppressing populations of many moth pests (Lance et al., 2016). The use of a synthetic codling moth sex pheromone (codlemone) for mating disruption has become a widely adopted, environmentally friendly component of integrated pest management against C. pomonella. Currently, it is used to suppress codling moth populations in over 160,000 ha of apple and pear orchards worldwide (Witzgall et al., 2008; Kadoić Balaško et al., 2020). Large-scale “area-wide” studies have shown significant reductions in fruit damage and conventional insecticide use (Bangels & Beliën, 2012; Kadoić Balaško et al., 2020). Despite higher upfront costs and variable efficacy under challenging environmental conditions (e.g., high pest densities, uneven pheromone distribution), mating disruption continues to expand globally due to its sustainability, precision, and ability to reduce insecticide inputs.

The mating disruption method offers advantages such as being species-specific and having no negative effects on non-target organisms, beneficial insects, human health, and the environment. Furthermore, it does not cause residue issues on fruits or lead to resistance development in insects, and is non-hazardous to farm workers. In addition, this method can be used in integrated pest management (IPM) and organic farming (Welter et al., 2005; Ahmed & Pfeiffer, 2010; Kamali, Koliaei & Taghadosi, 2017). However, this method also has some disadvantages. Its effectiveness depends on initial pest density, orchard size, distance from untreated orchards, and a uniform pheromone dispersion. High initial pest density, irregular canopy structures, steep terrain, and frequent wind exposure may compromise the success of the method. Furthermore, mating disruption demands strict and constant monitoring of the pest. Moreover, it is relatively expensive, especially compared to conventional insecticides, both in terms of pheromone dispenser costs and labor required (Ahmed & Pfeiffer, 2010; Kadoić Balaško et al., 2020). Mating disruption alone may not be a sufficient control method, especially when pest populations are very high. Therefore, an initial pesticide application may be required to reduce the pest population to a manageable level by mating disruption (Ahmed & Pfeiffer, 2010).

Various types of dispensers are used for mating disruption of C. pomonella. The components, contents, and application rates per hectare vary depending on the product. For example, Isomate-C Plus (190 mg (E,E)-8, 10-Dodecadienol) was applied at 1,000 pieces/ha (Sevinç et al., 2023), while Ecodian (18.75 g/ha of codlemone acetate and 6.25 g/ha of codlemone) was used at 900 m of wire per ha (Ferracini et al., 2021). Another dispenser, CheckMate® CM-F ((E,E)-8,10-dodecadien-1-ol, 14.3%), a sprayable microencapsulated sex pheromone formulation, was applied at 183 ml/ha, six times per season (Kutinkova et al., 2010), while Cidetrak, containing a unique combination of codling moth pheromone and kairomones that modify male and female behavior, was used at 20 pieces/ha (Palagacheva, Kutinkova & Dzhuvinov, 2021).

The aim of this study, conducted in the Beyşehir (Konya) district in 2023 and 2024, is to assess the efficacy of the mating disruption technique in controlling C. pomonella in commercial apple orchards. Additionally, this study will determine the adult population development of C. pomonella, the timing of the first adult emergence, the duration of adults’ active presence in nature, the number of generations, and the infestation rates in fruits.

Materials & Methods

Experimental orchards

The study was conducted in commercial apple orchards located in Beyşehir district (Konya, Türkiye) during the 2023 and 2024 apple growing seasons. Trials were conducted in a total of six apple orchards. Three of the six apple orchards were designated as pheromone-treated (PT) orchards, while the remaining three served as control (C) orchards. General characteristics of the orchards are shown in Table 1. The distances between orchards were 3.5 km for PT 1-C 1, 2.3 km for PT 2-C 2, and 3.6 km for PT 3-C 3. The trials were conducted in the same orchards in both years.

Table 1 General features of apple orchards used in mating disruption against Cydia pomonella in Konya (Türkiye).

Location	Area (ha)	Orchard age (years)	Number of trees (quantity)	Apple variety	Pheromone dispensers (quantity)	Altitude	Coordinates	
Pheromone-treated 1	1.6	13	800	Starking Golden Delicious	800	1,156	37°42′29″N 31°43′35″E	
Pheromone-treated 2	1.8	13	1,000	900	1157	37°43′05″N 31°43′17″E	
Pheromone-treated 3	1.3	14	750	650	1,154	37°42′48″N 31°42′58″E	
Control 1	1.8	11	1,000		1,163	37°44′06″N 31°42′16″E	
Control 2	2.5	30	1,000		1,156	37°42′58″N 31°43′15″E	
Control 3	2.5	30	1,000		1,157	37°42′52″N 31°43′24″E	

Obtaining climatic data

A HOBO data logger was installed in each apple orchard to obtain temperature and relative humidity values. Additionally, the obtained climate data were cross-checked with the data of meteorological stations of the Beyşehir District Directorate of Agriculture and Forestry.

Application of the mating disruption technique

Once the first adult was caught in Delta traps, ISOCOD-C (380 mg (E,E)-8,10-Dodecadienol, dodecanol, tetradecanol) pheromone dispensers (Shin-Etsu Chemical Co., Ltd., Tokyo, Japan) were hung at a dose of 500 pieces/ha, 1.5–1.8 m above the soil surface in four directions of the trees in the apple orchards where the mating disruption technique was applied. To reduce edge effect, twice as many pheromone dispensers were hung on each tree at the edge of the orchards. The efficacy of the treatment was determined by evaluating the number of males of C. pomonella captured in Delta traps.

Assessment of the mating disruption technique

Monitoring of Cydia pomonella

In each PT and C apple orchard, two Delta traps baited with synthetic sex pheromone (Russell IPM Ltd., Deeside, UK) were hung at 1.5–1.8 m above the soil surface, facing the southern direction of the trees, when the daily temperature reached 100 degree-days starting from January 1. The pheromone capsules in the Delta traps were replaced at regular intervals of 4–6 weeks, depending on the temperature, and the sticky cards were replaced when necessary. Delta traps were checked daily until the first males were captured. Afterwards, weekly checks were carried out to monitor male populations, and the number of males was recorded.

Sampling of infestation rate

When the fruits reached the walnut size in all the orchards, 10 fruits from 10 trees (a total of 100 fruits) randomly selected to represent each orchard were visually inspected weekly and the infested fruits were recorded. The number of infested fruits detected as a result of the controls was divided by the total number of fruits and the infestation rate was calculated. The infestation rate caused by C. pomonella was calculated using the following equation: Infestation rate%=Infested fruit numberInspected total fruit number×100.

Pesticide application

No insecticide application was made by us against C. pomonella in apple orchards. However, when necessary, the application of pesticides against C. pomonella and some other pests/diseases by the farmers was entirely at the initiative of the farmers. The study was carried out in commercial apple orchards completely under farmer conditions.

Data analysis

In both PT and C orchards, repeated measures ANOVA was used to analyze the mean number of C. pomonella males captured in Delta traps and the percentage of fruits infested by C. pomonella. These data were subjected to separate analysis on the basis of years. Tukey’s honestly significant difference (HSD) test was employed to evaluate the differences between means, using 0.01 error limits on the data. The mean percentage of fruit damage was transformed to arcsine before analysis. The statistical analyses were conducted utilizing SPSS 29.0 (IBM, 2023) software.

Results

Climatic data

HOBO devices were installed in each apple orchard to obtain climatic data. Additionally, data from the meteorological stations of the Beyşehir District Directorate of Agriculture and Forestry were gathered for comparison. The climatic data for the apple growing seasons of 2023 and 2024 in Beyşehir, Konya, are presented in Fig. 1.

Figure 1 The weekly average temperature (°C) and relative humidity (%) for the apple growing seasons of 2023 and 2024 in Beyşehir, Konya.

Population development of Cydia pomonella

The efficacy of the mating disruption technique against codling moth was assessed during the 2023 and 2024 apple-growing seasons in a total of six apple orchards: in three of these, ISOCOD-C pheromone dispensers were applied, and three were control orchards. In both apple-growing seasons, the number of male moths captured in Delta traps was significantly lower in the PT orchards compared to the C orchards. The population development and the number of male C. pomonella captured in both PT and C orchards are illustrated in Figs. 2 and 3.

Figure 2 The number of Cydia pomonella male captures in pheromone traps in apple orchards in 2023.

Figure 3 The number of Cydia pomonella male captures in pheromone traps in apple orchards in 2024.

In the first year of the study, Delta traps and ISOCOD-C pheromone dispensers were hung on trees on April 18, 2023. Due to heavy rains and low temperatures (Fig. 1), codling moths were not detected in the first four to five weeks after the traps were hung (Fig. 2). Considering all the apple orchards in the study, both PT and untreated, the first adults were caught in the traps on May 16, and the last ones on September 12. Therefore, it was determined that codling moth adults were active in nature for approximately four months. The pest population peaked twice during the 2023 production season. Hence, it was also found that the pest completed two generations in the Beyşehir district in 2023. The number of male moths in Delta traps in orchards where ISOCOD-C pheromone dispensers were applied was consistently lower than in C orchards during the 2023 production season. The capture of the first males in PT 1 orchard occurred on May 23 (5 adults/trap). In C 1, the first adults were caught on May 16, with the pest reaching its first peak on May 23 (42 adults/trap) and its second peak on August 8 (45 adults/trap). The first adults in Delta traps in PT 2 and C 2 were recorded on May 23 with 6 adults/trap and 63 adults/trap, respectively. The second peak in C 2 was reached on August 8 (48 adults/trap). The codling moth population showed similar development in PT 3 and C 3. In C 3, it formed two peaks on May 23 (57 adults/trap) and August 8 (44 adults/trap) (Fig. 2).

In the second year of the study, Delta traps and ISOCOD-C pheromone dispensers were placed on trees on April 21, 2024. Adult codling moths began to be caught in traps earlier due to reduced rainfall in April compared to the first year. Similar to the first-year trials, the pest population in PT orchards during the second year was consistently and significantly lower compared to the C orchards throughout the production season. In C 1, the first adults were caught on May 5 (9 adults/trap), with peaks observed twice on June 9 and July 14 at 62 adults/trap and 76 adults/trap, respectively. In C 2, the traps began to catch males from the second week following deployment, peaking three times on June 9, July 21, and August 18 with 64 adults/trap, 89 adults/trap, and 50 adults/trap, respectively. However, in the PT 2 orchard, the first adults were caught on May 19, five weeks after the traps were hung, and the total number of males captured in the whole production season was significantly lower than in C 2. In PT 3, the first adults were observed in traps seven weeks after the ISOCOD-C pheromone dispensers were hung, on June 2. In contrast, in C 3, the first adults were caught in traps on May 12, peaking twice on June 16 (72 adults/trap) and July 28 (69 adults/trap) (Fig. 3). Population observations were continued for two consecutive weeks after the harvest date (September 26 for 2023 and September 22 for 2024) in both PT and C orchards, until no adults were caught in the traps.

In the first year of the study, the mean number of males captured in traps in PT apple orchards was lower than the number of males captured in the C orchards, and a statistically significant difference was identified between them (F:7.815 and df:5, P < 0.01). The aforementioned observation was replicated in the second year of the study (F:11.384 and df: 5, P < 0.01). In 2023, the mean number of males caught in the traps in the PT orchards was 0.91 ± 0.18, while this was 11.38 ± 1.65 in the C orchards. Similarly, in 2024, the mean number of males caught in traps in PT and C orchards was 0.81 ± 0.19 and 19.60 ± 2.65, respectively (Table 2).

Table 2 Mean Cydia pomonella male captures in Delta traps in ISOCOD-C pheromone-treated and control orchards in 2023 and 2024 (Mean ± SE).

	2023	2024	
	Pheromone-treated	Control	Pheromone-treated	Control	
1	1.04 ± 0.34+a*	11.81 ± 2.78b	0.84 ± 0.33++a*	18.92 ± 4.39b	
2	0.92 ± 0.32a	11.31 ± 3.08b	0.96 ± 0.40a	25.60 ± 4.87b	
3	0.77 ± 0.28a	11.04 ± 2.78b	0.64 ± 0.27a	14.28 ± 4.39b	
Total+++	0.91 ± 0.18A	11.38 ± 1.64B	0.81 ± 0.19A	19.60 ± 2.65B	
Notes.

* In the same year, means followed by different letters in the same row are significantly different (P < 0.01).

+ F: 7.815 and df: 5 for the number of adults in 2023.

++ F: 11.384 and df: 5 for the number of adults in 2024.

F0.01;5;150 = 3.14.

+++ F: 84.505 and df: 1 for treatment.

F0.01;1;304 = 6.71.

Infestation rate of Cydia pomonella

In all orchards, fruit controls to determine the infestation rate were started when the fruit reached walnut size and continued weekly until harvest. In the first year of the study, the fruits reached walnut size on July 4. From this date onwards, 100 fruits randomly selected to represent each orchard were visually checked weekly and the number of infested fruits was recorded until September 26. During the whole production season, it was noted that the infestation rate in all three of the PT orchards was significantly lower than that observed in the C orchards. In addition, it was found that the infestation rate increased towards the harvest date. The infestation rate in PT 1 was between 0–3%, while this rate was determined as 4–25% in C 1. In PT 2, no infestation was detected in the first three weeks and the highest infestation rate recorded in 2023 was 3%, whereas it was recorded as 22% in C 2. The maximum infestation rates in PT 3 and C 3 were 4% and 23%, respectively (Fig. 4).

Figure 4 The infestation rates of Cydia pomonella in pheromone-treated and control apple orchards in 2023 and 2024.

In the second year of the study, the fruits reached walnut size on July 7, and fruit controls were carried out weekly until the harvest date, on September 22, 2024. The maximum infestation rate was 10% in PT 1, compared to 27% in C 1. The rate of infestation in PT 2 varied between 0 and 7%, whereas in C 2 it was between 8 and 30%. Similarly, it was noted that the infestation rate in PT 3 was lower compared to C 3. As in the first year, infestation rates were found to increase towards the harvest date.

The infestation rate of C. pomonella was consistently lower in PT orchards than in control ones for all orchards and both years. A statistically significant difference in infestation rates was identified between PT and C orchards (P < 0.01) in both years of the study (Table 3). In 2023, the mean infestation rate in the three PT orchards was 1.18 ±0.21, while this rate was 13.26 ± 1.08 in the C orchards. The mean infestation rate in the PT orchards (2.50 ± 0.43) increased in the second year compared to the first year of the study. However, the difference with the mean of the C orchards (15.33 ± 1.02) was found to be statistically significant.

Table 3 Mean percentage of fruit infested by Cydia pomonella in ISOCOD-C pheromone-treated and control orchards (Mean ± SE).

	Fruit infestation (%)	
	2023	2024	
	Pheromone-treated	Control	Pheromone-treated	Control	
1	1.15 ± 0.31+a*	14.54 ± 2.00b	3.42 ± 1.01++a	16.33 ± 1.83b	
2	1.15 ± 0.35a	12.69 ± 1.65b	2.42 ± 0.62a	15.08 ± 2.08b	
3	1.23 ± 0.44a	12.54 ± 2.05b	1.67 ± 0.48a	14.58 ± 1.45b	
Total+++	1.18 ± 0.21A	13.26 ± 1.08B	2.50 ± 0.43A	15.33 ± 1.02B	
Notes.

* In the same year, means followed by different letters in the same row are significantly different (P < 0.01).

+ F: 23.310 and df: 5 for parasitization rate (%) in 2023.

++ F: 26.264 and df: 5 for parasitization rate (%) in 2024.

F0.01;5;72 = 3.28.

+++ F: 247.547 and df: 1 for treatment.

F0.01;1;148 = 6.80.

Insecticide application

Insecticides were applied by farmers when required both in the PT and C orchards. In the initial year, insecticide application was carried out five times in the PT and nine times in the C orchards to control C. pomonella (Table 4). In the second year, the insecticide was applied six times in the PT and 11 times in the C orchards (Table 5). Therefore, it was determined that the total number of sprays for codling moth in PT orchards decreased by 44.4% and 45.4% in 2023 and 2024, respectively, compared to C.

Table 4 The number of sprayings, dates and active ingredients for Cydia pomonella in each pheromone-treated and control orchard in 2023.

Number of sprayings	Spraying date	Active ingredients	Pheromone-treated orchards	Control orchards	
1	23.05.2023	240 g/L Tau-fluvalinate	+	+	
2	30.05.2023	25 g/L Deltamethrin	+	+	
3	13.06.2023	100 g/L Alpha-cypermethrin	–	+	
4	27.06.2023	25 g/L Deltamethrin	–	+	
5	11.07.2023	50 g/L Lambda-cyhalothrin	+	+	
6	1.08.2023	25 g/L Deltamethrin	–	+	
7	8.08.2023	200 g/L Chlorantraniliprole	+	+	
8	22.08.2023	250 g/L Cypermethrin	+	+	
9	29.08.2023	50 g/L Lambda-cyhalothrin	–	+	
Total			5	9	

Table 5 The number of sprayings, dates and active ingredients for Cydia pomonella in each pheromone-treated and control orchard in 2024.

Number of sprayings	Spraying date	Active ingredients	Pheromone-treated orchards	Control orchards	
1	21.04.2024	240 g/L Tau-fluvalinate	+	+	
2	12.05.2024	25 g/L Deltamethrin	–	+	
3	21.05.2024	25 g/L Deltamethrin	–	+	
4	09.06.2024	100 g/L Alpha-cypermethrin	+	+	
5	16.06.2024	200 g/L Chlorantraniliprole	+	+	
6	30.06.2024	250 g/L Cypermethrin	–	+	
7	10.07.2024	50 g/L Lambda-cyhalothrin	–	+	
8	21. 07.2024	200 g/L Chlorantraniliprole	+	+	
9	28. 07.2024	25 g/L Deltamethrin	+	+	
10	11. 08.2024	25 g/L Deltamethrin	–	+	
11	25. 08.2024	250 g/L Cypermethrin	+	+	
Total			6	11	

Discussion

In the present study, it was determined that C. pomonella adults began flying at the end of April and the beginning of May. They remained active for 4–4.5 months and completed two generations per year in the Beyşehir district in 2023–2024. These results corroborate with the findings of previous studies. Çelik & Ünlü (2017) reported that C. pomonella produced 2–3 generations, and that the adult moths began to fly in the first week of May and remained active for 5 months in nature in Beyşehir district in 2014–2015. Aydoğan & Ünlü (2019) indicated that the first adult codling moth emergence was observed in the second week of May, remained active in nature for 5 months and completed 2–3 generations per year in Konya province in 2017 and 2018. Similarly, Icsık & Ünlü (2019) declared that the first adult emergence was observed in the first week of May. They were active in nature for 4–5 months, producing 2–3 generations per year in walnut orchards in Meram district (Konya).

ISOCOD-C pheromone dispensers were used for mating disruption against C. pomonella in this study. However, various pheromone dispensers such as Isomate-C Plus (Hepdurgun et al., 2001; Kutinkova et al., 2009; Madanat & Al-Antary, 2012; Walker et al., 2013; Demir & Kovanci, 2015; Sumedrea et al., 2015; İşci, Atasay & Kaymak, 2016; Horner et al., 2020; Öztürk & Hazır, 2020; Candan & Aslan, 2022; Sevinç et al., 2023), Ecodian (Angeli et al., 2007; Płuciennik, 2013; Ferracini et al., 2021), CheckMate (Kutinkova et al., 2010), Ginko (Kutinkova et al., 2020), CIDETRAK (Palagacheva, Kutinkova & Dzhuvinov, 2021), NoMate and Isomate 4Play (Walker et al., 2013) have been used against the pest in previous studies. Therefore, this is the first study to evaluate the efficacy of ISOCOD-C in the control of C. pomonella.

One of the values used to evaluate the efficacy of the mating disruption technique is the number of males captured in Delta traps in both PT and C orchards. There was a statistically significant difference (P < 0.01) in the number of C. pomonella adults caught in traps in the PT and C orchards in both years of the study. The mean number of adults caught in traps in PT orchards was 0.91 ± 0.18 and 0.81 ± 0.19, while it was 11.38 ± 1.64 and 19.60 ± 2.65 in C orchards. Therefore, it was concluded that mating disruption was effective in controlling C. pomonella in Beyşehir district apple orchards. Similar results were obtained in other studies carried out in Türkiye against the pest and Isomate-C Plus was used as a dispenser in all of these studies. Candan & Aslan (2022) reported that the total number of males caught in traps in the PT orchards (37 in the first year, 18 in the second year) was lower than in the C orchards (1,136 in the first year, 1,256 in the second year) in both years of the study in apple orchards in Kahramanmaraş province. Sevinç et al. (2023) conducted a study in the Isparta province and indicated that the number of males caught in traps in PT orchards never reached the economic threshold (six adults/trap) during the season. However, in C orchards, the population exceeded the economic threshold by 26 April 2021 and continued to increase weekly until the end of the season. Another study conducted in Isparta province reported that one adult was caught in the PT plot in the first year of the study, and none in the second year. Meanwhile, 117 and 85 adults were caught in the control plot in the first and second years, respectively (İşci, Atasay & Kaymak, 2016). A study conducted in walnut orchards found that the number of adults captured in PT orchards was lower than in C orchards for two consecutive years (Öztürk & Hazır, 2020).

The results of our study are also supported by studies conducted in other countries. Kutinkova et al. (2009); Kutinkova et al. (2010); Kutinkova et al. (2020) studied the effect of the mating disruption technique on C. pomonella in apple orchards by using different types of dispensers such as Isomate-C Plus, CheckMate and Ginko in Bulgaria. They indicated that pheromone dispensers totally inhibited the C. pomonella captures in pheromone traps hung in the trial plots in all studies. Ferracini et al. (2021) used the Ecodian pheromone dispenser in the mating disruption of Cydia spp. in northern Italy and found that the total number of males captured was significantly lower in PT plots compared to C plots. In Romania, Sumedrea et al. (2015) studied the mating disruption of Cydia spp. by using the pheromone dispenser Isomate-C. In the PT plot, the traps captured totally three adult moths, two of which were caught in May and one in June, and no moths were caught afterwards, while in the conventionally treated plot the number of captured moths was significantly higher. Walker et al. (2013) stated that the pheromone trap catches were reduced by 70%, from 40.1 adults/trap in the season before mating disruption was introduced to 11.7 adults/trap in 14 apple orchards in New Zealand.

Another value used to assess the efficacy of the mating disruption technique was the rate of infested fruits in both the PT and the C orchards. In this study, the codling moth infestation rate was consistently lower in PT orchards than in C orchards, and a statistically significant difference (P < 0.01) was detected for all orchards and both years. The mean infestation rates were determined as 1.18 ± 0.21–13.26 ± 1.08% and 2.50 ± 0.43–15.33 ± 1.02% in PT and C orchards, respectively. Previous studies support these results (Kutinkova et al., 2009; Kutinkova et al., 2010; İşci, Atasay & Kaymak, 2016; Öztürk & Hazır, 2020; Palagacheva, Kutinkova & Dzhuvinov, 2021). Conversely, Ferracini et al. (2021) were unable to obtain satisfactory data regarding fruit infestation, despite catching fewer adults in mating disruption plots. They reported that the larval infestation rate in fruits did not differ between mating disruption and control plots, except for one site.

The mean infestation rate in PT orchards never exceeded the economic threshold of 5% (Republic of Turkey Ministry of Agriculture and Forestry, 2019) during the two years of the study. Similarly, Öztürk & Hazır (2020) recorded the fruit infestation rate in the PT plot as below the economic threshold (5% and 4.8%) in both years of their study. Furthermore, Sevinç et al. (2023) found no evidence of infestation in the fruit trees in the trial orchard where pheromones were applied and the trees were sprayed three times against the first generation of the pest. However, İşci, Atasay & Kaymak (2016) found that the infestation rate in fruits was below the economic threshold in the initial year (2.38%) and above it in the subsequent year (13.50%). Moreover, Candan & Aslan (2022) determined the fruit infestation rate as above the economic threshold (9.07% and 8.38%) in both years of their study in the PT orchard.

A significant feature of the mating disruption technique is its capacity to reduce or even eliminate the negative effects of chemical control methods. There are studies in which no insecticide was used in orchards where the mating disruption technique was applied (İşci, Atasay & Kaymak, 2016; Öztürk & Hazır, 2020; Ferracini et al., 2021). However, the combination of mating disruption and a series of insecticide applications led to an increase in the effectiveness of C. pomonella control (Walker et al., 2013; Sumedrea et al., 2015). The use of the mating disruption technique alone has been found to be less effective in the control of C. pomonella. Madanat & Al-Antary (2012) found an infestation rate of 11.1% in the fruits in the area where only mating disruption was applied, compared to 4.2% in the area where mating disruption and four insecticide applications were used in the first year of their study against C. pomonella. In the second year, the rates were 8% and 5.8%, respectively. This study was carried out in commercial apple orchards completely under farmers’ conditions and insecticides were also applied when required in the PT orchards. However, interviews with the farmers showed that chemical control of codling moth was performed simultaneously in the apple orchards used in the study. Farmers in the same production area stayed in constant communication to coordinate codling moth control. Therefore, the fact that the study took place under farmers’ conditions did not introduce any variability and had no limiting effect on the results obtained. In the first year, five and nine sprayings were made against the pest in PT and C orchards, respectively, while in the second year, six and 11 sprayings were made, respectively. Although almost 50% less insecticide was applied in the mating disruption orchards, the infestation rate was found to be lower than in the C orchards in both years. These results congruently support the previous studies. Implementation of mating disruption in the control of the codling moth decreased the number of sprayings required to control this pest (Walker et al., 2013; Sumedrea et al., 2015; Kovanci, 2017). Sumedrea et al. (2015) applied insecticides 9 times to control the pest in mating disruption plots and 13 times in control plots for codling moth control. The respective numbers were 7 and 14 insecticide treatments in the subsequent year. In another study, the number of insecticide applications decreased from 5.9 to 3.7 after mating disruption was used (Walker et al., 2013). In the present study, the total number of sprays for codling moth in orchards using mating disruption decreased by 44.4% and 45.4% in 2023 and 2024, respectively, compared to the control. Similarly, Kovanci (2017) stated that the mating disruption technique used against codling moths in apple orchards reduced the total number of sprays for the apple pest complex by 40.7% and 56.6% in 2013 and 2014, respectively.

The comparatively elevated expense of mating disruption in comparison to conventional chemical control may act as a deterrent to its adoption by growers on a global scale (Kovanci, 2017). As in many other countries, the adoption of innovations by farmers in Türkiye is a lengthy process. Unfortunately, this also applies to the use of pheromone dispensers for both monitoring and control purposes in pest control. In both years of the study, mating disruption treatments resulted in a reduction in insecticide and machinery costs, whilst labor costs increased in comparison to conventional control. However, when all inputs, including insecticide application, labor, fuel, machinery, pheromone dispensers, and monitoring costs are considered, the cost of the mating disruption was found to be $376/ha and $304/ha more than the conventional control in 2023 in 2024, respectively. These results are supported by previous studies. Williamson et al. (1996) stated that the cost of codling moth mating disruption was $ 188.22/ha higher than that of conventional control methods. Similarly, Kovanci (2017) declared that the cost of mating disruption $ 193.70/ha higher than the conventional control in apple orchards. However, he stated that this situation may vary from year to year.

Conclusions

The results of the present study indicated that ISOCOD-C pheromone dispensers could successfully control the codling moth in apple orchards in Türkiye. Insecticide treatment was reduced by around 50% in PT orchards compared to C orchards. Furthermore, it was found that, despite the reduced use of insecticides, the infestation rate was significantly lower than in the control. Therefore, taking into account pesticide residue, resistance development, and the effects on natural enemies as well as on human and environmental health, it was concluded that the mating disruption technique has the potential to function effectively as an alternative to chemical control in the management of C. pomonella. However, in order for the technique to be successful, the ISOCOD-C pheromone dispensers should be hung in sufficient numbers as soon as the first adult flights are detected in the orchard. In addition, it is recommended to use more pheromone dispensers at the edges of the orchard to prevent the infiltration of gravid females from untreated neighboring orchards. In order to increase the success rate, the technique should be combined with appropriate insecticides when the population is high. When spraying, messages from the early warning system of official institutions should be taken into consideration, and unnecessary spraying should be avoided.

Supplemental Information

Supplemental Information 1 Raw data 2023: trap counts and infestation data

Supplemental Information 2 Raw data 2024: trap counts and infestation data

This study was summarized from Alperen Kutalmış’s master thesis entitled “Evaluation of the Efficacy of Mating Disruption Technique in the Control of Codling Moth (Cydia pomonella (L.) (Lepidoptera: Tortricidae)) in Apple Orchards of Beyşehir (Konya) District”.

Additional Information and Declarations

Competing Interests

Author Contributions

Data Availability

The authors declare there are no competing interests.

Alperen Kutalmış performed the experiments, analyzed the data, prepared figures and/or tables, and approved the final draft.

Ekrem Ögür conceived and designed the experiments, performed the experiments, analyzed the data, prepared figures and/or tables, authored or reviewed drafts of the article, and approved the final draft.

The following information was supplied regarding data availability:

The raw measurements are available in the Supplementary Files.

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
