# Peer review of "Control of Cydia pomonella (L.) (Lepidoptera: Tortricidae) in apple orchards using the mating disruption technique"

_PeerJ, doi:10.7717/peerj.20226_

## Round 0.1 · original submission · Major Revisions

· Academic Editor

Major Revisions

This study is the inaugural evaluation of ISOCOD-C dispensers for the management of codling moths in Türkiye, hence contributing innovation to the research.

However, the manuscript language needs to be polished. It is essential to correct typographical and encoding errors.

Comments of Reviewers should be taken into consideration.

**Language Note:** The review process has identified that the English language must be improved. PeerJ can provide language editing services - please contact us at [email protected] for pricing (be sure to provide your manuscript number and title). Alternatively, you should make your own arrangements to improve the language quality and provide details in your response letter. – PeerJ Staff

Reviewer 1 ·

Basic reporting

- The manuscript is generally well-written in scientific English, but there are occasional typographical and character encoding issues (e.g., “eûcacy” should be “efficacy”). A thorough language revision by a native speaker or professional editing service is strongly recommended.
- The article follows the conventional structure of scientific writing (Introduction, Methods, Results, Discussion, and Conclusion) and adheres to PeerJ guidelines.
- Figures and tables are informative and well-prepared. However, axis labeling and figure legends in some graphs should be clarified for improved readability.
- The literature cited is up-to-date and includes both national and international sources, strengthening the context of the study.

Experimental design

- The research question is clearly defined and relevant. The study aims to assess the efficacy of mating disruption using ISOCOD-C pheromone dispensers against Cydia pomonella under commercial apple orchard conditions in Türkiye.
- Field experiments were conducted in six orchards over two consecutive years, with three treated and three control orchards. The design is appropriate and replicable.
- One minor concern is that pest management (spraying) was left to the farmers’ discretion, which may introduce variability. This should be more explicitly addressed as a limitation.
- The methods are detailed and adequately described, including trap deployment, pheromone applications, monitoring protocols, and statistical analyses.

Validity of the findings

- The results are robust and statistically validated. The authors provide clear data on male moth captures, infestation rates, and insecticide reduction.
- Repeated measures ANOVA and Tukey HSD test are appropriate. The conclusions are directly supported by the findings.
- The study demonstrates that mating disruption significantly reduced male captures, fruit infestation, and insecticide use (by ~44–45%) over both years.
- This is the first study to evaluate ISOCOD-C dispensers for codling moth control in Türkiye, adding novelty to the research.

Additional comments

GENERAL COMMENTS
Strengths
- A well-structured, well-replicated, and relevant field study with practical implications for integrated pest management.
- Significant reduction in pest population and insecticide use is clearly demonstrated.
- The study aligns with sustainable agriculture goals.
Suggestions for Improvement
- Language polishing is essential to correct typographical and encoding errors.
- Some figure legends and axis labels could be improved for clarity.
- The discussion should more directly address potential variability due to farmer-managed spraying and edge effects from neighboring untreated orchards.
- A brief discussion of economic aspects or farmer adoption of pheromone dispensers would strengthen the applied value of the paper.

·

Basic reporting

The manuscript “Control of Cydia pomonella (L.) (Lepidoptera: Tortricidae) in apple orchards using the mating disruption technique” (#121731) reports the employs of ISOCOD-C pheromone dispensers to suppress the codling moth populations in Beyṣehir district (Konya) during the years 2023 and 2024. Results showed that ISOCOD-C pheromone dispensers suppressed capture of male moths and fruit infestation rate in the treated orchards. Moreover, the total number of insecticide sprays for codling moth control was decreased by 44.4% and 45.4% in 2023 and 2024, respectively.

Experimental design

1.In this study, ISOCOD-C (380 mg ((E,E)-8,10-Dodecadienol, dodecanol, tetradecanol) pheromone dispensers were hung at a dose of 500 pieces/ha. What is the basis for choosing the dose of 500 pieces/ha of the ISOCOD-C in this study? Besides this dose, did the authors compare other doses (less than, and greater than 500 pieces/ha) ?
2.In this study, the authors hanged dispensers at 1.5-1.8 m above the soil surface. In fact, many people do it this way, but this is not the optimal suspension height. Generally, it should be hung at a height 20% below the top of the tree canopy.
3.The use of mating disruption for pest control has certain requirements for the size of the orchard. The orchard should not be too small to avoid the insignificant disorientation effect of the sex pheromones on pests. Judging from the longitude and latitude information of the experimental orchards in Table 1, there are distance intervals among these orchards. However, it is recommended that the authors still need to clearly state in Materials & Methods how much distance is between the control orchards and the pheromone-treated orchards.

Validity of the findings

The findings in this study offers valuable insights that could be integrated into area-wide pest management strategies for the suppression of this agricultural pest.

Additional comments

1.In Introduction, the authors need to systematically expound on the current global situation of using mating disruption technology to control codling moth, including the advantages and disadvantages of this technology, applicable scenarios (such as population density), the components and contents of dispensers pheromones used in different countries, and whether other pest control measures are needed as supplements.
2.Additionally, in other countries and regions that use mating disruption to control codling moth, what are the main components and contents of the pheromone in the dispensers they use? What is the hanging density of the dispensers? These details should be clearly stated in the Introduction.
3. Generally, synthetic sex pheromone is subject to specificity issues. Besides being able to trap target insects, some non-target insects can also be captured. During the monitoring process, did the authors also trap other pests, such as Grapholitha molesta, etc. Because the pheromone components of other insects closely related to the Cydia pomonella are very similar to those of Cydia pomonella, this to some extent interferes with the mating disruption effect of the Cydia pomonella.

---

## Round 0.2 · accepted · Accept

· Academic Editor

Accept

Thank you for addressing all of the reviewers' comments. The manuscript was proofread and edited by a native English speaker which make the manuscript ready for publication.

·

Basic reporting

The authors have addressed all my concerns.

Experimental design

--

Validity of the findings

--

Additional comments

--